# Crossing Phylums: Butterfly Wing as a Natural Perfusable Three-Dimensional (3D) Bioconstruct for Bone Tissue Engineering

**DOI:** 10.3390/jfb13020068

**Published:** 2022-06-01

**Authors:** Fatemeh Mostofi, Marzieh Mostofi, Behnaz Niroomand, Saadi Hosseini, Atefeh Alipour, Shahin Homaeigohar, Javad Mohammadi, Mohammad Ali Shokrgozar, Hosein Shahsavarani

**Affiliations:** 1Department of Cell and Molecular Biology, Faculty of Life Science and Biotechnology, Shahid Beheshti University, Tehran 19839-69411, Iran; fatemehmostofi79@gmail.com; 2Laboratory of Regenerative Medicine and Biomedical Innovations, Pasteur Institute of Iran, National Cell Bank, Tehran 13169-43551, Iran; marziehmostofi2000@gmail.com (M.M.); behnaz.niroomand@protonmail.com (B.N.); sadihosseini95@gmail.com (S.H.); mashokrgozar@pasteur.ac.ir (M.A.S.); 3Department of Cell and Molecular Biology, Faculty of Biological Sciences, Kharazmi University, Tehran 31979-37551, Iran; 4School of Medicine, Shahid Beheshti University of Medical Sciences, Tehran 19839-63113, Iran; 5Department of Nanobiotechnology, Pasteur Institute of Iran, Tehran 13169-43551, Iran; 6School of Science and Engineering, University of Dundee, Dundee DD1 4HN, UK; 7Department of Life Science, Faculty of New Science and Technology, University of Tehran, Tehran 14399-57131, Iran; javad.mohammadi@ut.ac.ir

**Keywords:** butterfly wings, chitin-based scaffold, tissue engineering, osteoblasts, mesenchymal stromal cells

## Abstract

Despite the advent of promising technologies in tissue engineering, finding a biomimetic 3D bio-construct capable of enhancing cell attachment, maintenance, and function is still a challenge in producing tailorable scaffolds for bone regeneration. Here, osteostimulatory effects of the butterfly wings as a naturally porous and non-toxic chitinous scaffold on mesenchymal stromal cells are assessed. The topographical characterization of the butterfly wings implied their ability to mimic bone tissue microenvironment, whereas their regenerative potential was validated after a 14-day cell culture. In vivo analysis showed that the scaffold induced no major inflammatory response in Wistar rats. Topographical features of the bioconstruct upregulated the osteogenic genes, including *COL1A1*, *ALP*, *BGLAP*, *SPP1*, *SP7*, and *AML3* in differentiated cells compared to the cells cultured in the culture plate. However, butterfly wings were shown to provide a biomimetic microstructure and proper bone regenerative capacity through a unique combination of various structural and material properties. Therefore, this novel platform can be confidently recommended for bone tissue engineering applications.

## 1. Introduction

Bones are dynamic structures in the human body with limited self-repair capability against physical injuries or traumas [1]. As a result, regeneration of damaged bone tissues is one of the most significant concerns for researchers in the biomedical field [2]. Given the high importance of bone health, the World Health Organization (WHO) has named the first decade of the 21st century (2000–2010) as the decade of bone and joint [3].

Some common causes of generation of bone defects include motorcar accidents, sport-related injuries, osteoporosis, osteonecrosis, and primary or metastatic malignancies of bones, which impose a significant burden on healthcare systems across the world [4].

The currently applied method, autograft, deals with the repair of the lesion site via harvesting intact bone from the iliac crest and grafting it to the damaged area. Despite several advantages of this method, complications such as morbidity and possible infection of the donor site increase the risk of bleeding and complexity of the operation. Additionally, the limited amount of harvesting material challenges the management of the injury. Therefore, the available approaches cannot solve the problem entirely and impose some adverse complications on the patients [5,6].

Bone tissue engineering combines material and life science principles to fabricate implantable bioconstructs to replace the damaged bone [7]. This technique is based on three main components including scaffold biomaterials, stem cells, and growth factors [8]. Various studies have been performed to find a biodegradable and mechanically compatible scaffold that is capable of inducing osteogenic differentiation. Biocompatible scaffolds that provide adult stem cells with proper conditions for growth and differentiation are better mimics of natural extracellular matrix (ECM) [9].

Tissue engineering (TE) scaffolds are meant to create an appropriate niche for cell migration and proliferation, tissue reconstruction, nutrition and growth factors’ delivery, and waste removal from cells [9]. To achieve these goals, TE scaffolds should feature a proper surface for cell adhesion, high porosity, a high specific surface area, negligible toxicity and immunogenicity, biocompatibility, biodegradability, and optimum mechanical properties [9,10].

Researchers in the field of tissue engineering have studied many natural and synthetic scaffold materials to find the most appropriate ones for tissue reconstruction. The electrospun scaffold fabricated via PHB with incorporated rGO or PANi is one example of synthetic bioconstructs [11]. In this regard, due to various reasons such as immunogenicity and biocompatibility, nature-derived scaffold materials have received more attention than synthetic ones. Decellularization of natural scaffolds excludes the antigenic content, thus eliminating immunogenicity. In contrast, the antigenic content of synthetic scaffolds is usually unknown and depends on the type of material they are made up of [9,12]. The biocompatibility of natural scaffolds is also higher than synthetic ones. Moreover, synthetic scaffolds are poorly biodegradable compared to the natural scaffolds and typically release toxic byproducts during degradation [13].

In recent years, production of natural scaffolds derived from plant and animal sources has attracted the attention of TE researchers [14,15,16,17,18,19,20,21]. For instance, Salehi et al. used cabbage as a scaffold material for osteogenic differentiation [22]. In another study, they examined osteogenic differentiation on spinach leaves [23]. Despite such biological merit, complex approaches needed for the decellularization of plant-derived scaffolds restrict their capacity for further changes and manipulation. For example, a decellularized plant-derived scaffold cannot provide the kidney tissue with a proper chemical gradient [24]. Cellulose-based scaffolds are not biodegradable in vivo, and thus cannot be replaced with newly formed natural tissues [25]. In general, only a few plant species can be regarded appropriate to develop scaffolds supporting cell growth and differentiation [26].

Nowadays, the chitin-based scaffolds are widely used due to their low toxicity, biodegradability, biocompatibility, and renewability [27]. The chemical structure of chitin is a polymer of N-acetyl-d-glucosamine (NAG) linked by beta-glycosidic bond [28]. Chitin is the most abundant carbohydrate in nature after cellulose and a natural cationic polysaccharide similar to glycosaminoglycans, the main element of the ECM. They can be degraded into small molecular amino and polysaccharides in vivo, which are non-toxic and easily absorbed. It has been confirmed that the molecular structure, surface morphology, and porosity of chitin scaffolds can influence the biological behavior of seed cells, including cytoskeleton arrangement, proliferation, adhesion, and differentiation [7]. Moreover, they enhance immune response, offer antibacterial activity, and accelerate the tissue repair process [29,30,31]. Insects are one of the most significant chitin sources [32,33] and have been studied for development of natural TE scaffolds. For example, Elbaz et al. synthesized chitin TE scaffolds based on wings of three different butterfly species. Such scaffolds induced promising cell viability and proliferation [15,17]. In another study, Wang et al. employed the wings of *Morpho menelaus* and *Papilio ulysses telegonus* as TE scaffolds with improved cells survival and grow [34]. Moreover, chitin extracted from shrimp shells, red sea demosponges, mushrooms, and cockroaches has been proved to fulfill the criteria of a suitable scaffold [35,36,37,38,39,40].

We intentionally chose *Papilio demoleus* among several species of butterflies due to its chitin abundancy as well as its appropriate wing surface morphology that enables the cells to adhere without the need to decellularization. On the other hand, these butterflies are regarded as a citrus pest and are abundant during spring and summer. As a result, the applied strategy is not only cost-effective but also eco-friendly. In this article, we hypothesized that *P. demoleus* wings could provide an appropriate scaffold for osteogenic differentiation, according to their biocompatible, porous, non-cytotoxic chitin-based structure. The main objectives of this study are to (1) examine surface properties, polymer structure, and biocompatibility of the wings of *P. demoleus* butterfly, (2) apply new and more achievable methods to improve the wing’s surface properties, (3) demonstrate its efficacy in directing osteogenic differentiation of Ad-MSCs, (4) introduce the wings of *P. demoleus* as a novel, eco-friendly, and cost-efficient source of chitin-based scaffold for cell replication and differentiation.

## 2. Materials and Methods

### 2.1. Sample Collection

Butterflies of *P. demoleus* species were collected from citrus gardens located in southwestern Iran in spring. The samples were stored in a closed container in a dry atmosphere while being transferred to the laboratory. The butterfly species had been already approved by the Iranian Research Institute of Plant Protection (IRIPP).

### 2.2. Materials

Hydrochloric acid (HCl) (37%), sodium hydroxide (NaOH), chloroform (CHCl_3_), and ethanol (96%) were purchased from Merck (Darmstadt, Germany). Phosphate-buffered saline (PBS) and Collagenase type I were obtained from FUJIFILM Wako Pure Chemical Corporation (Osaka, Japan). Tetrazolium dye MTT 3-(4,5-dimethylthiazol-2-yl)-2,5-diphenyltetrazolium bromide and Dulbecco’s modified eagle Medium-F12 were purchased from Sigma-Aldrich (Burlington, VT, USA). Pen-Strep solution and fetal bovine serum (FBS) were obtained from Gibco (New York, NY, USA).

### 2.3. Scaffold Preparation

#### 2.3.1. Physico-Chemical Treatment of the Butterfly Wings

The butterfly wings were carefully separated using a cutter, and their delicate structure was maintained properly. The wings were subsequently washed with distilled water and later were discolored via physical and chemical treatments. First, the scales were carefully removed from the wing surface by a brush, and then the wings were soaked in 1 M HCl solution for 4 h followed by a 15-min immersion in 2 M NaOH solution at 55 °C. The samples were eventually washed three times with distilled water for 5 min each time.

#### 2.3.2. Hydrophilizing the Butterfly Wings

The physico-chemically treated wings were submerged in chloroform for 2 h to raise their hydrophilicity [41]. The samples were then plasma-treated for 40 s in an expanded tabletop plasma cleaner AC/DC input 230 V AC (PDC-002/PDC-FMG-2) with the power of 150 W.

### 2.4. Scanning Electron Microscopy

Scanning electron microscopy (SEM) (KYKY, EM3200) was performed to characterize the morphology of the scaffold surface as well as the cells adhered on the scaffold. Previously, the samples were sputter-coated with gold. The obtained images were analyzed using MIP (Microstructure Image Processing) software to evaluate the surface physical structure, porosity, and distance between pores.

### 2.5. Atomic Force Microscopy (AFM)

To image the topographical features of the scaffold, atomic force microscopy (Veeco Auto Probe CP research) was performed. Using AFM, root mean square (Rq) and average roughness (Ra) were measured on 5 µm × 5 µm slices of the scaffold. Additionally, the obtained results were processed to create three-dimensional images.

### 2.6. Fourier Transform Infrared (FTIR) Analysis

ATR-FTIR (spectrum 400 Perkin Elmer) analysis was used to determine the surface chemistry of the butterfly wing scaffolds. To do so, the samples were IR irradiated in the spectral range of 450–4000 cm^−1^ at 22 °C and under an air humidity of 20%.

### 2.7. Contact Angle Measurement

A water contact angle test was performed on the surface-treated and untreated wings to validate the efficacy of the applied treatments on their hydrophilicity. To do so, the static contact angle was measured using a JIKAN CAG-20 device by depositing a water droplet on the sample surface (1 cm^2^) at room temperature and measuring the water contact angle after 10 s.

### 2.8. Surface Area and Pore Volume Determination

Brunauer–Emmett–Teller (BET) (BELSORP MINI II) analysis was used to determine the specific surface area, pore volume, and pore size of the scaffold [42,43]. The analysis was carried out by applying nitrogen adsorption isotherms within bath temperature range of −196 to 15 °C. To prepare the samples, they were placed in a degassing machine (BEL PREP VAC II) at a temperature of 120 °C for 2 h.

### 2.9. In Vivo Animal Tests

To validate the biocompatibility of the wing scaffolds, they were implanted on the skin of rats. Twelve Wistar rats, weighing 263 g on average, were purchased from the Animal Breeding Laboratory and Care Center of Shahid Beheshti University, Iran. The rats were kept in separate cages and were submitted to the same dietary treatment with free access to drinkable water. The rats were eventually anesthetized using chloral hydrate (0.365 g/mL, Merck Co., Darmstadt, Germany), and circular excision wounds (0.5 cm in diameter) were made on shaved areas on their back to allow subcutaneous implantation of the wing scaffolds.

To identify any inflammation response upon implantation of the wing scaffolds, four separate blood samples were taken weekly for three successive weeks. The systemic inflammation markers including inflammatory factors interleukin-6 (*IL-6*) and tumor necrosis factor-alpha (*TNF-alpha*) were measured in the blood samples using ELISA. For histopathological analysis, four tissue sections were obtained on days 1, 7, 14, and 21 under general anesthesia. The as-prepared sections were fixed in 10% paraformaldehyde (Wako Company, Osaka, Japan) and stained with hematoxylin-eosin. Moreover, some tissue samples were treated to extract RNA using a DENAzist Column RNA isolation kit (DENAzist Asia Co., Mashhad, Iran). Real-Time RT-PCR was performed to detect any changes in the expression levels of *IL-6*. The cDNA was synthesized by the cDNA synthesis kit (Parstous Company, Mashhad, Iran). Furthermore, the primer sequences were designed by CLCwork and Oligo7 software (Table 1). The histopathological analysis was meant to rule in/out inflammation by detecting some clues, such as fibroblast aggregation and neutrophil or monocyte infiltration.

### 2.10. Cell Culture

Following the chemical treatment and hydrophilization of the scaffolds, they were sterilized using UV irradiation for 40 min and incubation in 70% ethanol for 2 min. Then, they were washed three times with PBS 1× and 2.5% penicillin/streptomycin for 15 min. The samples were later transferred to a 24-well microplate containing cell culture medium and incubated for 12 h in 5% CO_2_, 37 °C, and 90% humidity. In this study, human mesenchymal stem cells (MSCs) harvested from adipose tissue stem cells (Prepared by National Cell Bank, Pasteur Institute of Iran, Tehran, Iran) were monitored by a CD14, CD45, CD34, CD73, CD90, and CD105 marker, as similarly shown by Azadian et al. [44]. The cells were seeded on the scaffolds already placed in 24-well tissue culture plates. A cell density of 10^4^ cells per well was used for this purpose and allowed to adhere to the scaffold surface. Subsequently, 500 μL of DMEM F12 medium supplemented with 10% FBS, 2 mM glutamate, 1% penicillin-streptomycin, and 15 mM sodium bicarbonate were added to the wells. The scaffold-cell co-cultures were incubated at 37 °C and 5% CO_2_. Later, differentiation of MSCs to osteoblasts was investigated at 7, 14 and 21-day intervals. Moreover, control cells were cultured with an osteogenic medium supplemented with dexamethasone (12 μM) (Sigma Aldrich), ascorbic acid (50 μg/mL) (FUJIFILM Wako Pure Chemicals Corp., Richmond, VA, USA), and β-glycerophosphate (10 μM) (Sigma-Aldrich) on tissue culture plates (TCPS).

### 2.11. Cytotoxicity

Cytotoxicity and cell viability of the scaffolds were quantified using the MTT assay. Human foreskin fibroblasts (HFFs) (Prepared by National Cell Bank, Pasteur Institute of Iran, Tehran, Iran) were seeded on the scaffold sterilized by UV radiation and 70% ethanol. After several incubation periods of 1, 3, 5, and 7 days, supernatants were aspirated out, and 10 µL MTT solution (50 mg/10 mL PBS) was added to each well containing the scaffolds. Then, 90 µL of culture medium was added to each well, and incubation continued for 4 h. After that, MTT and culture medium were removed. Later, formazan was dissolved in 100 µL DMSO (Wako Co., Osaka, Japan), and well plates were incubated at 37 °C in a shaker incubator for 20–30 min. Finally, optical density was recorded at λ ~ 590 nm using an ELISA reader (Benchmark, Bio-Rad, Hercules, CA, USA). The cell adhesion capability and cell proliferation evaluated by MTT assay and Dapi staining (Section 2.12.1), Saso2 cell line (Prepared by National Cell Bank, Pasteur Institute of Iran, Tehran, Iran) were seeded on the wing scaffolds and TCPS control substrates. After incubation for 1, 7, 14, and 21 days, cell proliferation and cell adhesion were determined according to the protocol instructed by the MTT manufacturer.

### 2.12. Biocompatibility Assays

#### 2.12.1. DAPI Staining

To investigate the cell viability and cell adhesion on the scaffolds, the cell nuclei were stained with DAPI (4′,6-Diamidino-2-phenylindole) (Wako Co., Osaka, Japan). To do so, on the 14th day after the cell culture, the cells present on the scaffold were fixed using 4% paraformaldehyde, were washed three times with water, and then exposed to the DAPI reagent (5 μg/mL) for 10 min. Eventually, the samples were washed twice with PBS and imaged at different magnifications by a fluorescence microscope (Bell Engineering INV100-FL, Monza (MB), Italy) with a UV filter.

#### 2.12.2. Field Emission Scanning Electron Microscope (FE-SEM)

After 2 weeks of scaffold-cell co-culture, FE- SEM imaging was perfromed to monitor the cells growth on the scaffold. For this reason, the samples were fixed with 4% paraformaldehyde (Wako Co., Osaka, Japan) for 30 min and washed with PBS three times. After that, the scaffolds were dehydrated using gradients of ethanol (20% to 96%). Finally, the scaffolds were gold-coated at 15 mA for 3 min using a Hitachi E-1010 ion-sputtering device.

### 2.13. Alizarin Red-S Staining

Alizarin Red (AR) staining was used to demonstrate the extent of calcium deposits formed by Ad-MSCs cultured on the scaffold. The cultured cells were fixed with 4% paraformaldehyde and were subsequently stained with 2% (*w*/*v*) Alizarin Red S (Bioidea Co., Tehran, Iran) for 5 min. The dye was dissolved in distilled water (pH 7.2), and the samples were later washed twice with PBS. Finally, the results were inspected by a phase-contrast light microscope.

### 2.14. Alkaline Phosphatase (ALP) Activity Measurement

The ALP activity of the cells was measured using an Alkaline phosphatase kit (Biorexfars Co., Shiraz, Iran) after 7- and 14-days cell culture. First, the total protein content of the samples was extracted manually using the RIPA solution. Afterward, ALP activity was determined according to the manufacturer’s kit. The unit value of the enzyme was normalized to the total protein content. Eventually, the optical absorbance of the samples was measured at λ~405 nm by a microplate reader (BioTek Epoch, Santa Clara, CA, USA).

### 2.15. Real-Time RT-PCR

The relative expression of the genes involved in differentiation of MSCs into osteoblasts, including collagen type I alpha 1 chain (*COL1A1*), alkaline phosphatase (*ALP*), bone gamma-carboxyglutamate (*BGLAP*; *Osteocalcin*), secreted phosphoprotein 1 (*SPP1*; *Osteopontin*), SP7 transcription factor 7 (*SP7*), and runt-related transcription factor 2 (*RUNX2*; *AML3*) as well as the *GAPDH* reference gene, was quantified by Real-Time-PCR (Applied Biosystems PCR System, Life Technologies, Carlsbad, CA, USA). Additionally, the primer sequences were designed by the CLCwork and Oligo7 software (Table 1).

Total RNA content was determined after 2 weeks of culturing the cells on the scaffold using the DENAzist Column RNA Isolation Kit (DENAzist Asia Co., Mashhad, Iran). RNA quantity and quality were characterized using NanoDrop (Agilent, Milan, Italy) and electrophoresis (1.8% agarose gel), respectively. The single-strand cDNA synthesis was performed using the cDNA synthesis kit (Parstous Company), according to the manufacturer’s instructions. Fold changes in gene expression were determined using the comparative Ct method (ΔΔCt) with normalization to the housekeeping gene, *GAPDH*. The obtained data were analyzed by the Relative expression software tool (REST) and GraphPad Prism 8 software.

### 2.16. Immunocytochemistry (ICC) Test

The immunocytochemistry (ICC) test was performed using Osteocalcin antibody to characterize osteogenic differentiation. For this purpose, at first, the cells were incubated in PBST (PBS + Triton x100 (0.2%) (Merck)) for 5 min at room temperature to improve the cell migration. After washing with PBS, the well plates were incubated with PBSTw for one hour at 37 °C. After additional washing with PBS (three times), the cells were kept in PBSTW (Bovine serum albumin (BSA) 1% + Glycine) at 37 °C for 30 min. The Human anti-rabbit primary osteocalcin antibody (ab93876, Abcam, Cambridge, UK) was added in PBSTW and incubated at 4 °C for 24 h (the osteocalcin antibody dilution rate of 1:2000). After washing with PBS (three times), BSA (1%) was added in dark, incubated for 1 h, and washed three times (each time for 5 min). After washing with PBS, supplemented with 0.1% Tween-20, the cells were incubated by secondary antibody (for 2 h) and then washed by PBS. The cells were observed by fluorescent microscope (Bell Engineering INV100-FL, Monza (MB), Italy) using the 430–560 nm filter.

### 2.17. Statistical Analysis

All experiments were repeated three times, and the obtained data were expressed as mean ± SD. Data analysis was performed by GraphPad Prism 8 and compared using two-way analysis of variance (ANOVA) with a repetition t-test. A *p*-value of less than 0.05 was considered significant.

## 3. Results

### 3.1. Scaffold Preparation

After physical removal of the scales, the wings became transparent and less hydrophobic, and thus more prone to chemical treatment. Moreover, the chemical treatment removed the residual scales and made the wings completely transparent. Figure 1a,b show the butterfly wing before and after physico-chemical treatment, respectively. Furthermore, according to FTIR results, using these chemicals to prepare the scaffold has not significantly affected its structure and composition because they have not been exposed to chemicals for a long time.

### 3.2. Scaffold Characterization

#### 3.2.1. Morphology and Topography of the Scaffold

SEM images (Figure 1c,d) demonstrate a regular pattern of the unidirectional ridges on the wing surface. The morphological analysis of the chemically treated wings by the MIP software indicates the existence of pores, with an average space of 75 μm between them. Such a porous structure enables the cells to notably adhere and grow on it.

#### 3.2.2. Atomic Force Microscopy (AFM) Analysis

Statistical analysis of the surface is crucial for quantifying the surface features. Therefore, the scaffold roughness was assessed using atomic force microscopy. The results of the surface height distribution analysis showed that the average roughness (Ra) is 27.87 nm, which falls in the nanoscale roughness. The surface skewness, which is an indication for deviation in the surface height distribution from normal distribution, was +0.209. This positive deviation from the normal or Gaussian surface height distribution indicates that the peaks are predominant on the surface, which increases the surface available for the cells to adhere to the surface. The results also indicate an opportune microscale roughness that implies the promising potential of the scaffold to encourage cell adhesion and activation (Figure 2a).

#### 3.2.3. Fourier Transform Infrared (FTIR) Spectra Analysis

One of the main factors governing the cell attachment and proliferation onto a substrate is the chemical cue comprising of surface functionalities, which further controls some physical cues such as wetting and surface hydrophilicity. The chemical analysis of the substrate provides insights into its structure and the functionalities available. As shown in Figure 2b, N-H and O-H stretching bands were found at 3350 cm^−1^; presence of a broad and asymmetric double band in the wavenumber range of 3170–3580 cm^−1^ is an indication of overlapped structural -OH and -NH bands. Symmetric CH_3_ and asymmetric CH_2_ stretching bands at 2963 cm^−1^, CH stretching band at 2883 cm^−1^, and C=O of amide I band at 1659 cm^−1^ in the ATR-FTIR spectrum results. Moreover, the bands at 1539 cm^−1^, 1449 cm^−1^, and 1379 cm^−1^ are related to the N-H of amide II, CH_2_ bending and CH_3_ deformation, and CH bending and symmetric CH_3_ deformation bands, respectively. Furthermore, bands at 1308 cm^−1^, 1156 cm^−1^, 1037 cm^−1^, and 743 cm^−1^ correspond to the amide III and CH_2_ wagging, asymmetric bridge O_2_ stretching, CO stretching, and NH out-of-plane bending bands, respectively. The obtained bonds match the standard chemical bonds of chitin-based polymers; accordingly, our scaffold predominantly consists of chitin [45,46].

#### 3.2.4. Water Contact Angle

The effect of the physico-chemical treatment on hydrophilicity of the butterfly wings was monitored using water contact angle. As shown in Figure 2c,d, the water contact angle for the intact and the physically treated wings were measured to be 135° and 97°, respectively. Moreover, the chemically treated wing showed a complete spread of water droplet on the scaffold after 5 s of the sprinkling it (Figure 2e). Therefore, it can confidently be said that the physico-chemical treatment successfully increased the hydrophilicity of the butterfly wings, thus creating a proper platform for cell adhesion and proliferation.

#### 3.2.5. Brunauer–Emmett–Teller (BET) Analysis

The porous structure of the *P. demoleus* wing was characterized by BET. As tabulated in Table 2, the surface area (a_s,BET_), total pore volume, and average pore diameter are 20.073 m^2^ g^−1^, 0.1461 cm^3^ g^−1^, and 29.119 nm, respectively.

### 3.3. In Vivo Experiments

Skin erythema and swelling, fever, weight loss, and diet habit changes were monitored for 21 days. Skin reactions were expected to emerge, considering the invasive nature of the applied procedure. However, no exudative discharge and no dramatic swelling were observed. The body temperature of the test group showed a slight increase on days 1 to 3 after implantation; however, a high-grade fever related to infection was absent. A weight loss of 5–10 g was detected, and the food consumption of the rats remained unchanged. In the presence of the scaffold, inflammatory factors such as Interleukin-6 (IL-6) and tumor necrosis factor-alpha (TNF-α) were at a higher level compared to the control on the 1st day, but thereafter they declined. This behavior implies the mildness of the systemic response, originated from the invasive procedure (Table 3). In concordance with ELISA outcomes, RT-PCR results of IL-6 expression level show the same pattern as blood samples (Figure 3a). Figure 3b–g, and the tissue hematoxylin-eosin staining images taken on days 7 and 21 (Figure 3h–k), show that skin infection decreases over time, and that the wounds are healed normally.

### 3.4. Cell Proliferation and Viability

As verified by a MTT assay, the cells maintain their optimum viability in the presence of the wing scaffold and properly proliferate (Figure 4a). The percentage of cell growth inhibition increases at a relatively low rate until the 7th day (Figure 4c). Moreover, the proliferation of the Saso2 cell line is significantly high until the 21st day (Figure 4b).

### 3.5. DAPI Staining

A spreading, viable cell population is evident from the analysis of the fluorescent microscopy micrographs. Blue spots present the active areas of cell nucleus viability, confirming the non-cytotoxicity of the scaffold (Figure 4e,f).

### 3.6. Cell Morphology

Micrographs of FE-SEM revealed adhesion of cultured cells with their cytoplasmic process adhered to the surface. Micrographs on days 14 and 21 (Figure 5a–c) are evident of the osteogenic morphogenesis of the MSCs.

### 3.7. Calcium Deposition

Calcium content as a sign of osteogenic differentiation of mesenchymal stem cells was investigated by Alizarin red staining. The results of this staining showed significant calcium deposition in Ad-MSCs cultured on the scaffold compared to Ad-MSCs cultured on the TCPS (Figure 5d–g).

### 3.8. Alkaline Phosphatase Activity

ALP activity was measured to investigate the osteogenic differentiation of MSCs on the scaffolds. Results show that ALP activity of MSCs cultured on the scaffold was significantly greater than MSCs and MSCs with standard osteogenic differentiation treatment cultured on TCPS and gradually increased until the 21st day (Figure 5h).

### 3.9. Gene Expression Analysis

As verified via RT-qPCR, the expression level of *COL1A1*, *ALP*, and *SP7* genes were upregulated on day 21 compared to day 14. While the expression of *AML3*, *BGLAP*, and *SPP1* genes increased until the 14th day, it decreased on day 21, indicating the differentiation of MSCs into mature osteoblasts. Moreover, the expression of *ALP* gene in the cells cultured on the wing scaffold significantly raised compared to those cultured on TCPS (control+ sample), Figure 6. This behavior reflects the better differentiation conditions available in the presence of the scaffold.

### 3.10. Immunocytochemistry (ICC) Test

The results of the immunocytochemistry assay showed that osteocalcin protein expirations on the surface of differentiation cells on the scaffold confirm the efficiency of the scaffold in differentiated MSCs to osteoblasts (Figure 7).

## 4. Discussion

The Scaly cover of butterfly wings has an anisotropic arrangement in micro-, submicro-, and nano-scales [47]. It is anticipated that the remaining regular array of protuberances after descaling can guide cell attachment and orientation. In this study, we selected the *P.domeleus* species because of its abundance as an agricultural pest, a green source, and, the most important, anisotropic wing structure [48]. Descaling simply with a brush and then removing possible cytotoxic constitutes of the wing surface using the two-step chemical treatment were feasible ways of increasing biocompatibility of the wings.

The FTIR survey has confirmed that chitin is the basic and dominant background material of the butterfly wing. Chitin scaffolds have been found to influence osteogenesis-related signaling pathways and stimulate mineralization to facilitate bone regeneration. Chitin scaffolds can provide a proper template for the deposition of calcium phosphate crystals to the mineralization process of the organic matrix in the human body. Moreover, they possess biological properties required for a functional scaffold, such as biocompatibility, biodegradability, low immunogenicity, and chemical stability [7].

Butterfly wing-based scaffolds do not demand high-tech equipment or complex protocols for decellularization, so they are easily treatable, and we treated wings in the briefest way possible. However, complex decellularization approaches are required in the case of cellulose-based scaffolds derived from plants. For example, in our previous studies, we established multi-step approaches to decellularize cabbage leaves and onion to obtain a cellulose-based scaffold for osteogenic differentiation [22,49].

Moreover, the preparation of synthetic scaffolds is time-consuming, as, for example, in the construction of two kinds of porous ceramics, a novel sintered porous hydroxyapatite, and a porous β-tricalcium phosphate (β-TCP), as well as a collagen-phosphosphoryn sponge as scaffolds for bone cell culture, which took a relatively time-consuming process to prepare [50]. On the other hand, the osteogenic activity of synthetic polymers might not meet the needs of bone tissue repair, such as in the case of poly (L-lactide) (PLLA)-based porous scaffolds, which have been extensively fabricated through 3D printing; their osteogenic activity still does not meet the needs of bone tissue repair [51].

It is observed that hydrophilicity increase cell-material interaction, cell adhesion, proliferation, infiltration, and osteogenic differentiation. It also contributes to better mineralization and Ca^2+^ deposition [52]. Extreme hydrophobicity of the wings of *P. domeleus* butterfly disrupts the ability of cells to attach and interact with the scaffold [31]. Hydrophobicity of *P. demoleus* wings was significantly overcome by the two-step chemical and subsequent chloroform and plasma treatment, causing a decrease from 135° to complete absorption in water contact angle. Moreover, it is evident that physical removal of scales as the natural water barrier reduces the angle (135° to 97°).

Surface roughness affects cell response biology and, in particular ranges of average roughness (Ra), can enhance cell adhesion and proliferation. Surface roughness at the nanoscale mimics the surface of natural tissue and at the microscale increases the bioactivity through the absorption of proteins. Our scaffold exhibits enough roughness at both scales based on the AFM results, making it an appropriate surface for cell activation. In one study, it has been shown that the nanoscale Ra of the silicon surface, between 0 and 64 nm (not above), can positively promote neural cell adherence. Considering suitable cell attachment based on FE-SEM imaging and DAPI staining in our study, the same range can be generalized to Ad-MSCs on the chitin surface. Fluorescent microscopy of DAPI-stained cell-laden wings also reveals an almost uniform distribution of cells on the surface, which validates successful cell attachment [53,54,55]. DAPI staining images in concordance with MTT results showed acceptable viability of cells after 21 days. It demonstrates that the scaffold does not produce cytotoxic byproducts and is safe for cell culture.

In vivo experiments on Wistar rats showed that the scaffold is tolerated well by the rats. Mild fever and local allergic reactions were induced, but systemic inflammatory response and agitated behavior were not observed during the follow-up period. Therefore, the scaffold did not induce infection or exudative response, which might indicate anti-microbial and anti-inflammatory traits of it and also confirmed biocompatibility and non-toxicity of the scaffold.

Porosity is the key feature to successful bone scaffolding. Material without pores does not serve osteogenic differentiation [56]. This feature is essential for cell adhesion, nourishment, and oxygenation [57,58]. The analysis of BET results confirms the presence of pores throughout the scaffold with an average diameter of 29 nm.

Osteogenesis-related activities were surveyed using Alizarin Red staining, immunocytochemistry, ALP enzyme assay, and RT-PCR. Ca^2+^ ions are necessary elements in the formation of hydroxyapatite in bones, so they are essential components of osseous ECM. We assessed the scaffold’s capacity of mineralization by Alizarin Red staining. Alkaline phosphatase activity is also another determinant in bone ossification. The results of both analyses show the differentiation of MSCs into osteoblasts. Osteocalcin is a calcium-binding peptide that is secreted by osteoblasts. As a non-collagen protein, it is the most abundant protein of organic bone matrix after collagen [59]. Its presence on the scaffold has been demonstrated by ICC assay; therefore, the scaffold is an appropriate surface for osteoblasts to synthesize and secrete one of the most important proteins of the non-ossifying bony matrix. Similar to osteocalcin, alkaline phosphatase (ALP) ectoenzyme is a specific marker for bone formation. ALP activation is crucial to the osteoid formation and mineralization of organic matrix [60]. In line with the ALP assay result, cells cultured on the scaffold activated the ALP significantly more than cells cultured on TCPS on day 21. From day 7 to 21, the ALP activity was maintained and upregulated, indicating a constructive cell-material interaction.

The expression of osteogenesis-related genes is substantial to confirm that bone formation is happening in the proposed scaffold. For instance, the *AML3* gene is thought to play a role in the differentiation process of MSCs into mature osteoblasts since it regulates major signaling pathways related to osteogenesis such as the hedgehog, Fgf, and Wnt signaling. Activation of a major signaling pathway such as Wnt/β-catenin signal cascade leads to upregulation of other essential transcription factors to osteogenic differentiation such as *Runx2* [61,62]. *COL1A1* and *ALP* are early markers of osteogenic differentiation. *COL1A1* is crucial to bone phenotyping. Based on RT-PCR results, both show a sustained increasing level of expression from day 7 to 21. *BGLAP* is the most abundant structural molecule in bone after *COL1A1* and is one of the few molecules specific to this tissue [63]. The expression pattern of *BGLAP* is decreasing from day 14 to 21, similar to the positive control, but it shows slightly higher levels of expression than the positive control. Therefore, the scaffold can provide cells with enough facilities to direct their differentiation through osteogenic-related gene expression.

In similar studies, Elbaz et al. reported that NIH-3T3 fibroblast cells were attached and survived well on the scaffolds obtained from the wings of three different butterfly species, including *Morpho menelaus*, *Papilio ulysses telegonus*, and *Ornithoptera croesus lydius* [17]. They have reported in another study that the HEPG2 hepatoma cell line can be survived on butterfly wings [15]. However, the capability of butterfly wings for the culture of any human stem cells and directing their fate has not yet been examined. The present study investigated the cell attachment, survival rates, and mechanism of regulating mesenchymal stem cells differentiation that have less desire to adhere surfaces and are more sensitive to growth requirements. The data presented here confirmed the effects of topographical stimuli resulting from butterfly wings’ microenvironmental structure to effectively induce osteogenesis in human adipose-derived mesenchymal stem cells.

Furthermore, in a recent study, the wings of *M. menelaus* and *Papilio ulysses telegonus* were used as cultures for neural cells, which have been shown to provide enough facilities for cell growth and cell orientation. They have inspected mature Schwan cells and neural ganglia and did not check the effect of physicochemical cues of butterfly wings in guiding stem cell fate [34]. Finding an appropriate culture for the growth, proliferation, and differentiation of these cells is challenging for biologists. Therefore, our study confirms the capacity of the scaffold for these purposes is more substantial than mentioned studies.

Despite many advances in scaffold fabrication methods, various clinical trials should examine their efficacy and safety as implantable grafts. The ability to establish homeostasis, vasculature link with adjacent tissue, and regeneration are vital needs of a tissue, which future studies should evaluate proposed scaffolds [23,64].

## 5. Conclusions

This study introduces the extraction methodology and the biological applicability of the wings of *P. demoleus* as the renewable source for developing an appropriate scaffold. In vitro, biological studies confirmed that simply treated *P. domeleus* butterfly wings can be used as biocompatible, biodegradable, and nontoxic scaffolds for mesenchymal stem cell culture and proliferation. Due to the proper surface morphology revealed by the SEM results and cell adhesion, viability, and differentiation proved by the assays mentioned earlier, the inherent ability of the scaffold to promote bone formation has been established in our study. In conclusion, the scaffold could be used in bone tissue engineering as the optimal candidate for the treatment of bone damages, along with its easy availability and cost-effectiveness.

## Figures and Tables

**Figure 1 jfb-13-00068-f001:**
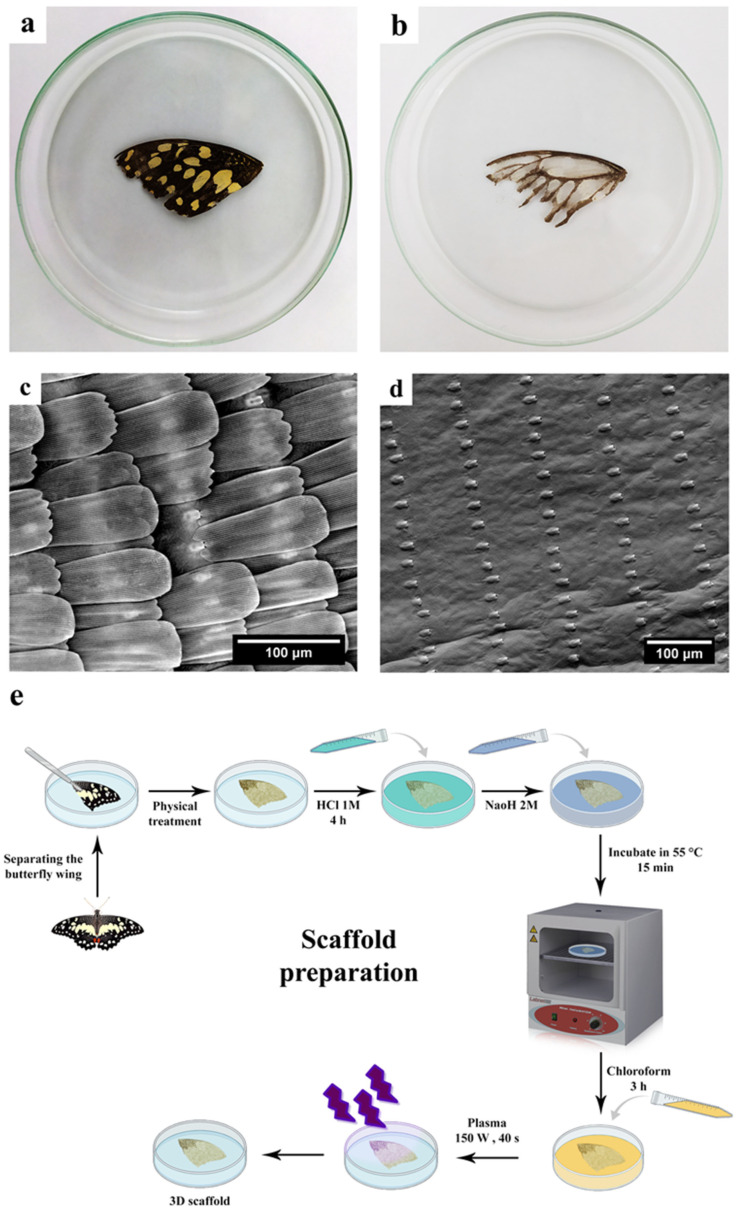
Images of *P. demoleus* wing (**a**) control sample, (**b**) sample with physical and chemical treatment, (**c**,**d**) Scanning electron microscope (SEM) image from the surface of *P. demoleus* wing before and after physico-chemical treatment, respectively (800× and 500×, respectively), and (**e**) Supplementary figure of scaffold preparation.

**Figure 2 jfb-13-00068-f002:**
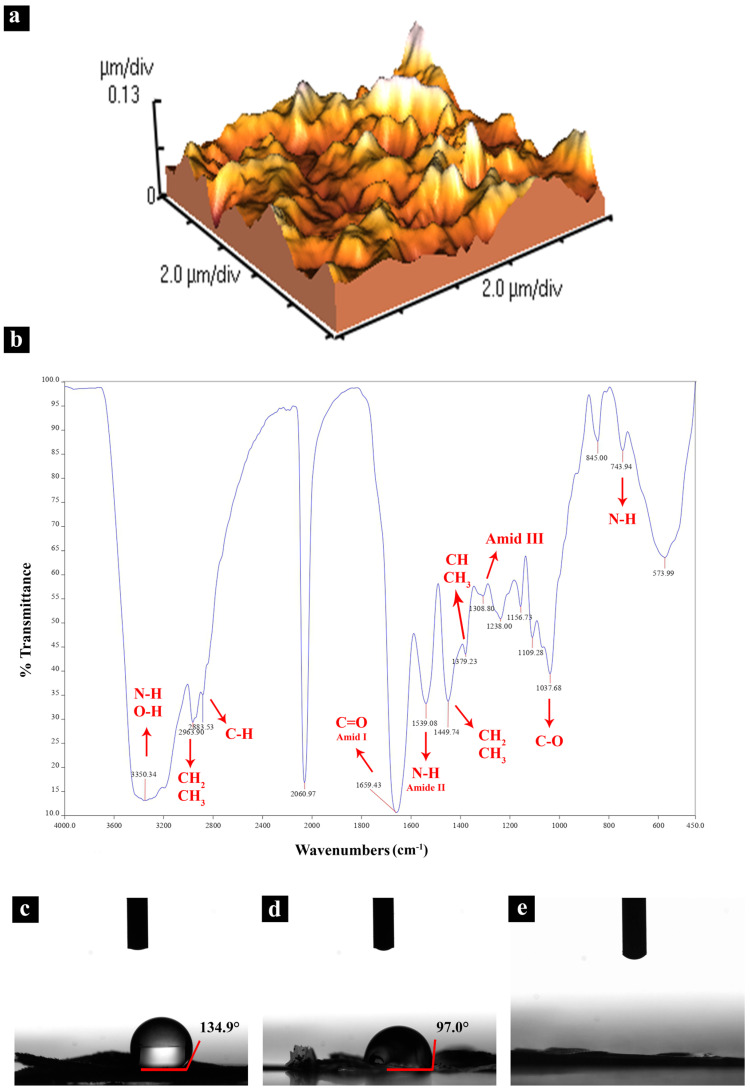
The results of scaffold characterization (**a**) surface roughness analysis by atomic force microscope (AFM), (**b**) the results of Fourier-transform infrared (FTIR) analysis of the butterfly wing, (**c**) contact angle before treatment, (**d**) after physical treatment, and (**e**) after physical and chemical treatment with HCL, NaOH, chloroform, and plasma.

**Figure 3 jfb-13-00068-f003:**
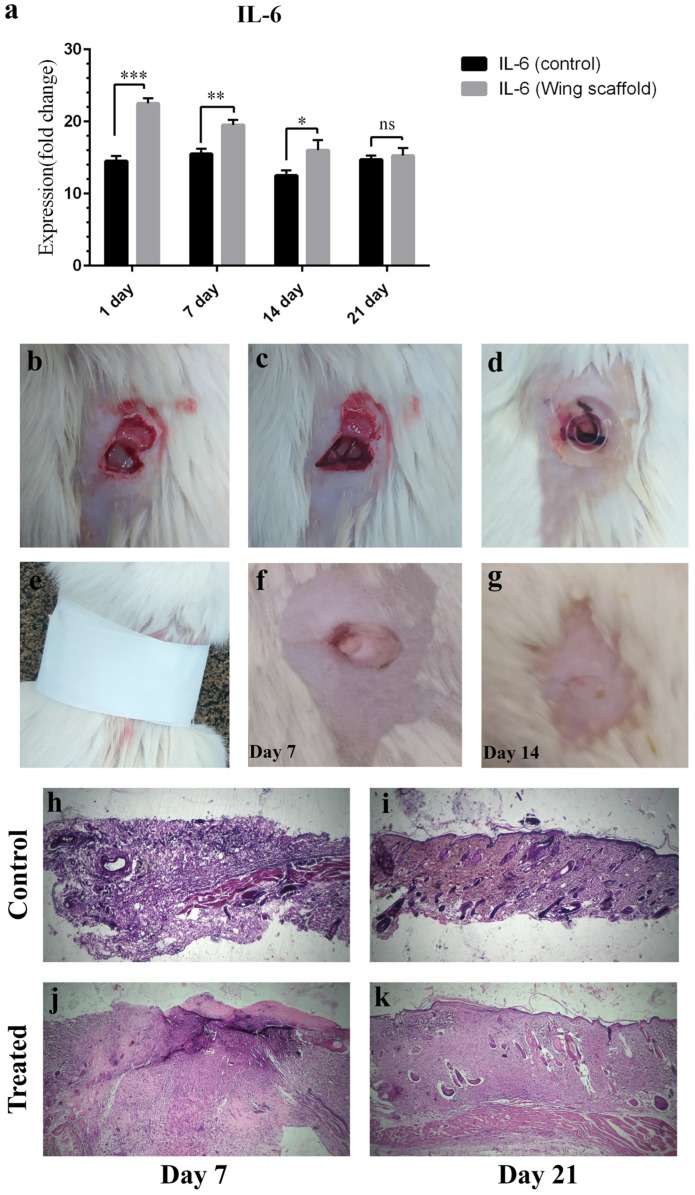
In vivo experiments images (**a**) the expression level of Interleukin-6 (IL-6) gene. Significant differences between groups are indicated with a star sign (*p* < 0.05). (**b**–**e**) Making wound, implantation of the scaffold, (**f**,**g**) wound healing steps on days 7 and 14, and (**h**–**k**) tissue hematoxylin-eosin staining of control and treated sample on days 7 and 21.

**Figure 4 jfb-13-00068-f004:**
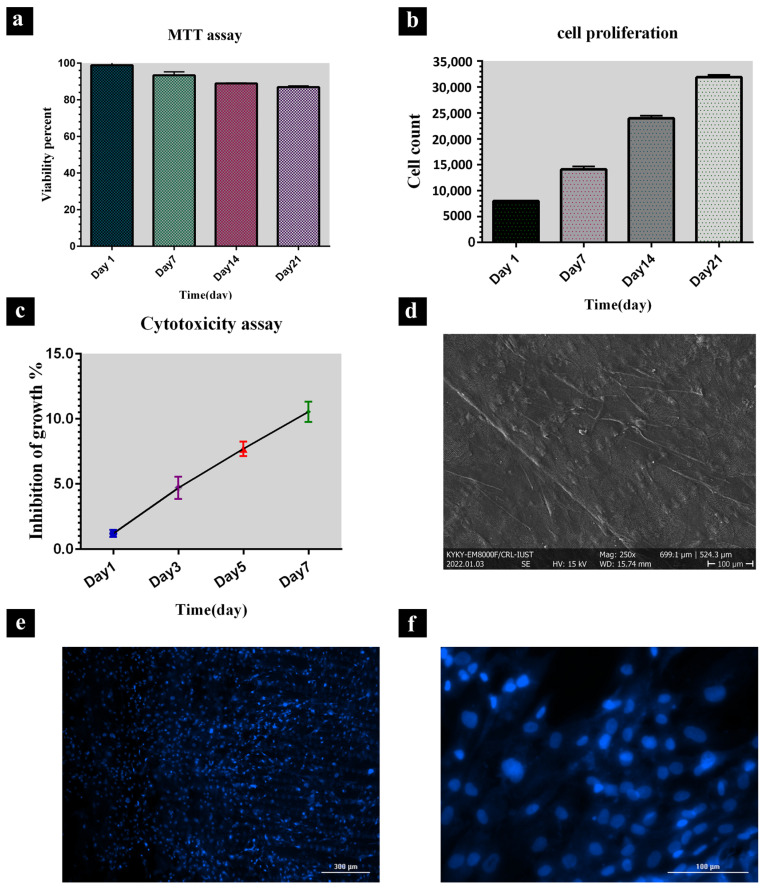
Biocompatibility assays (**a**) results obtained from MTT assay when HFFs were cultured on butterfly wing during days 1, 7, 14, and 21 (**b**) proliferation of Saso2 cell line when they were cultured on butterfly wing during 3 weeks, (**c**) percentage of Ad-MSCs growth inhibition when cultured on butterfly wing during days 1, 3, 5 and 7 (**d**) Field Emission Scanning Electron Microscopes (FE-SEM) image and (**e**,**f**) 4′,6-Diamidino-2-phenylindole (DAPI) staining image (with different magnifications) of mesenchymal stem cells on the *P. demoleus* wing show viability and cell adhesion.

**Figure 5 jfb-13-00068-f005:**
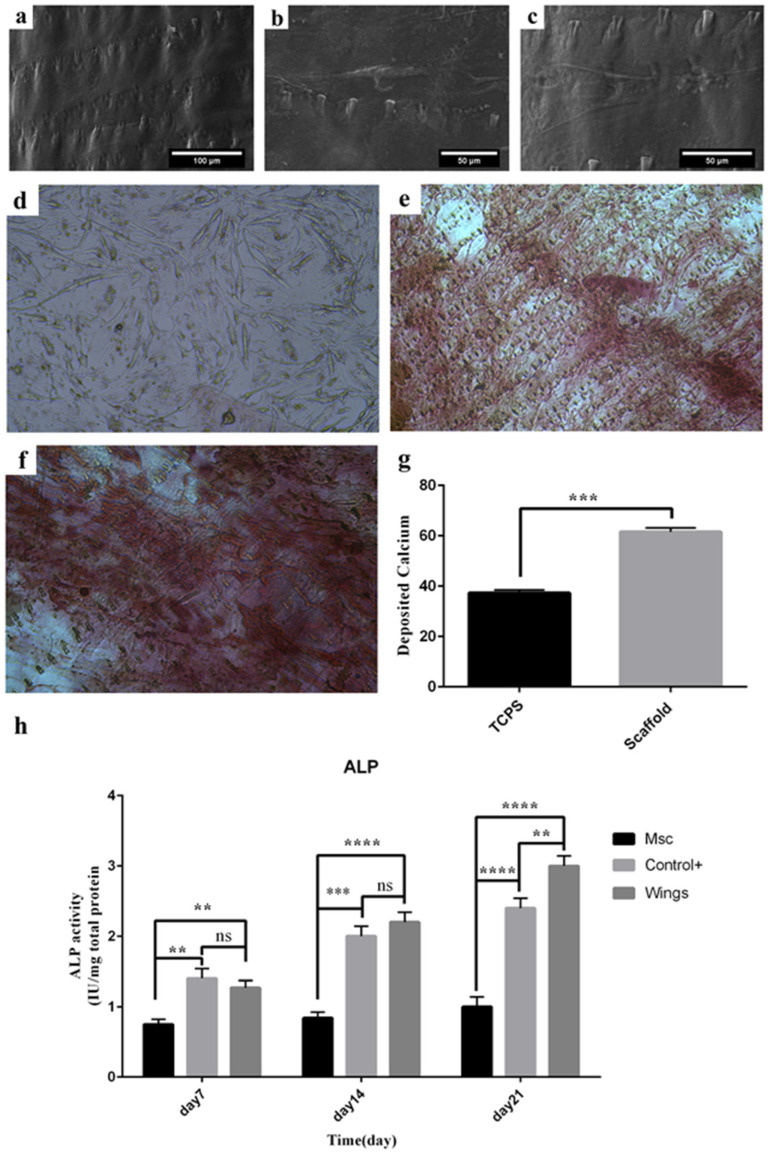
Differentiation assays (**a**) Field Emission Scanning Electron Microscopes (FE-SEM) from the surface of *P. demoleus* wing before and (**b**) after 14 days, (**c**) 21 days of Ad-MSC cultures. (**d**) Images captured by phase-contrast light microscope from Alizarin-Red staining 21 days after culture. Mineralization staining of Ad-MSCs when culture on the TCPS (control sample) and (**e**,**f**) on the scaffold with different magnifications (10× and 20×, respectively), (**g**) quantified calcium deposition based on Alizarin red staining (Significant differences between groups are indicated with the star sign), and (**h**) Alkaline phosphatase (ALP) activity of MSCs (control), MSCs with standard osteogenic differentiation treatment cultured on TCPS (control+), and MSCs cultured on the scaffold, on days 7, 14 and 21. (ns: not significant, ** (*p* < 0.01), *** (*p* < 0.001), **** (*p* < 0.0001).

**Figure 6 jfb-13-00068-f006:**
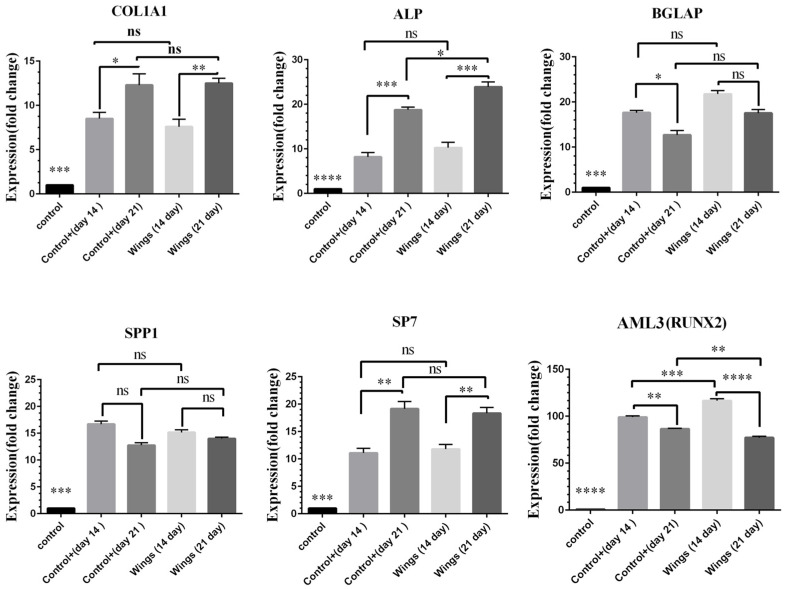
The expression level of osteogenic-related genes (*COL1A1*, *ALP*, *BGLAP*, *SPP1*, *SP7*, and *AML3*) in MSCs when cultured on the TCPS without treatment (control), with treatment (control+), and when cultured on the scaffold (wing) on days 14 and 21, respectively. (ns: not significant, * (*p* < 0.05), ** (*p* < 0.01), *** (*p* < 0.001), **** (*p* < 0.0001).

**Figure 7 jfb-13-00068-f007:**
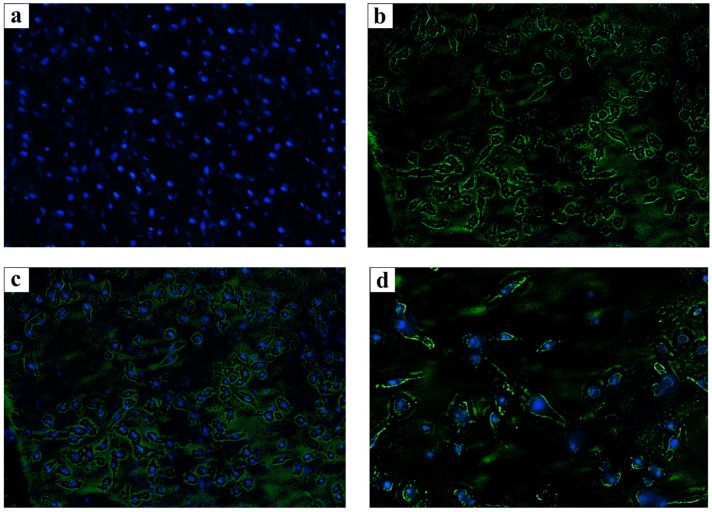
Immunostaining of cells on the *P. demoleus* wing (**a**) 4′,6-Diamidino-2-phenylindole (DAPI) staining, (**b**) Fluorescein Isothiocyanate (FITC) staining, and (**c**,**d**) the merged field with different magnifications (10× and 40×, respectively).

**Table 1 jfb-13-00068-t001:** The sequence of the primers used to RT-PCR of genes related to osteogenic differentiation and inflammatory factors.

Genes	Forward Primer Sequence (5′→3′)	Reverse Primer Sequence (5′→3′)	Product Length (bp)
COL1A1	CATCTCCCCTTCGTTTTTGAC	CCAAATCCGATGTTTCTGCTG	121
ALP	ACCATTCCCACGTCTTCACAT	GACATTCTCTCGTTCACCGC	161
BGLAP	TCACACTCCTCGCCCTATTG	CTCTTCACTACCTCGCTGCC	133
SPP1	GAGGTGATGTCCTCGTCTGTAG	CACATATGATGGCCGAGGTG	111
SP7	TACCCCATCTCCCTTGACTG	GCTGCAAGCTCTCCATAACC	110
AML3	AGATGATGACACTGCCACCTC	GGGATGAAATGCTTGGGAACT	125
GAPDH	ACCCACTCCTCCACCTTTG	CTCTTGTGCTCTTGCTGGG	178
r-IL-6	TCCTACCCCAACTTCCAATGC	GTTTGCCGAGTAGACCTCAT	138
r-GAPDH	GGCAAGTTCAACGGCACAG	CGCCAGTAGACTCCACGAC	142

**Table 2 jfb-13-00068-t002:** The results of Brunauer–Emmett–Teller (BET) test.

Scaffold	a_s,BET_ (m^2^ g^−1^)	Total Pore Volume (cm^3^ g^−1^)	Average Pore Diameter (nm)
Butterfly wing	20.073	0.1461	29.119

**Table 3 jfb-13-00068-t003:** In vivo experiments: the levels of inflammatory factors Interleukin-6 (IL-6) and Tumor necrosis factor-alpha (TNF-α) detected by ELISA.

Sample	TNF-α (pg/mL)				IL-6 (pg/mL)			
	Day1	Day7	Day14	Day21	Day1	Day7	Day14	Day21
Control ^a^	25.1	27.1	26.3	25.7	15.4	17.2	16.1	15.9
Scaffold ^b^	37.1	33.8	29.9	26.5	25.1	24.7	19.9	17.1
Significant (*p* ≤ 0.05)	****	***	*	ns	***	***	*	ns

IL-6 = interleukin-6, TNF-α = tumor necrosis factor-alpha. (a) Without scaffold, (b) Butterfly wing (ns: not significant, * (*p* < 0.05), *** (*p* < 0.001), **** (*p* < 0.0001).

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
