# Peer review of "Crossing Phylums: Butterfly Wing as a Natural Perfusable Three-Dimensional (3D) Bioconstruct for Bone Tissue Engineering"

_jfb, 2022, doi:10.3390/jfb13020068_

Round 1
Reviewer 1 Report
The manuscript entitled "Butterfly wings as a natural perfusable three-dimensional (3D) bio-construct for bone tissue engineering" is a fascinating thought. The thought is an innovation in developing scaffolds made from natural ingredients. The findings is expected for providing a biomimetic microstructure and proper bone regenerative capacity through a unique combination of various structural and material properties. Therefore, this novel platform can be confidently recommended for bone tissue engineering applications.
Some points to note are:
- Regarding the benefits and guarantee of sustainability from the availability of scaffolds derived from butterfly wings, can naturally butterfly wings be obtained all year around? How to standardize the butterfly wing material that can be used to make a scaffold that can support the proliferation of mesenchymal stem cells and osteoconductive in bone regeneration?
- What details are the 3D scaffold characteristics needed to be a means for mesenchymal stem cell growth and proliferation?
- This study ignores using a control scaffold that is commonly used in the medical application of bone fractures that are already available in the market, such as scaffolds based on hydroxyapatite or Bovine Hydroxyapatite. Thus, this data cannot provide a sufficient illustration regarding the advantages of the butterfly wing as a scaffold in bone regenerative.
- In the abstract, it was written, "In vivo analysis showed that the scaffold induced no allergic response in Wistar rats." At the same time, the examination carried out was the expression of TNFa and IL6, which are inflammatory markers related to the wound healing process. In contrast, no direct evidence points to "no allergenic" materials such as the determination of IgE expression or levels in experimental animals.
- Can the in vitro study approach conclude that the simply treated P. domeleus butterfly wings can produce biocompatible and biodegradable scaffolds for mesenchymal stem cell culture and proliferation?
- Examination of osteocalcin in mesenchymal stem cell culture was not accompanied by a control group, as shown in Figure 7. Therefore, these data cannot be interpreted ultimately. Likewise, a higher resolution is needed so that visual evaluation can be carried out quickly.
Author Response
Dear Dr. Prof. Dr. Francesco Puoci
Editor-in-Chief of the Journal of Functional Biomaterials
We would like to express our deep gratitude to you, editors and reviewers for providing us with the insight and direction needed to complete our submitted manuscript under the title of “Crossing phylums: Butterfly wings as a natural perfusable three-dimensional (3D) bio construct for bone tissue engineering”.
We have carefully taken your comments and questions into the consideration for preparing revised manuscript. Please kindly find the revised version of our manuscript in which the revised parts are indicated in red color in revised manuscript. However, to some extent we would like to emphasize the response to your questions and major comments in this reply sheet as follows.
I hope our responses to your comments meet your expectations to initiate its reviewing process. This manuscript has been checked by language editing services prior to submission.
Please kindly let us know if you or reviewer evaluate that this revision needs further revision. We would be very happy if our work could be acceptable for publication in internationally renowned Journal of Functional Biomaterials.
Thanks in advance,
Hosein Shahsavarani, PhD
Reviewers' comments:
Reviewer #1
The manuscript entitled "Butterfly wings as a natural perfusable three-dimensional (3D) bio-construct for bone tissue engineering" is a fascinating thought. The thought is an innovation in developing scaffolds made from natural ingredients. The findings is expected for providing a biomimetic microstructure and proper bone regenerative capacity through a unique combination of various structural and material properties. Therefore, this novel platform can be confidently recommended for bone tissue engineering applications.
Response to the general comments of reviewer #1: We would like to express our thanks to you for all your suggestions that are needed to be revised to fulfil the requirements to accept in JFB journal. We hope our responses to your comments meet your expectations. Please kindly let us know if you evaluate that this revision is enough mature to be accepted for publication or needs further revision.
Some points to note are:
- Regarding the benefits and guarantee of sustainability from the availability of scaffolds derived from butterfly wings, can naturally butterfly wings be obtained all year around? How to standardize the butterfly wing material that can be used to make a scaffold that can support the proliferation of mesenchymal stem cells and osteoconductive in bone regeneration?
Response to the Q1: Our study first aimed to clarify capability of the natural bioinspired biomaterials such as butterfly wings based on their chemical and mechanical properties to direct stem cell fate. In response to your question, we would like to note that the P.demoleus butterfly is available in most months of the year. Overall, the most favorable time period for this butterfly is July to December, which is half of the year. Besides this fact, the butterfly can be grown in simulated environmental conditions, which requires further deep study. On the other hand, our study shows that the chitin-based structure and the topographical features of the P.demoleus wings are appropriate for osteoinduction. To standardize this scaffold for large-scale application, we suggest using 3D bio-printers programmed based on the wing’s topographical features and its main background material- chitin. The SEM and AFM analyses provide visual and statistical data for designing a 3D surface- patterned scaffold using bio-printers.
- What details are the 3D scaffold characteristics needed to be a means for mesenchymal stem cell growth and proliferation?
Scaffolds, at the best, possess features that include biocompatibility, biodegradability, good cell adhesion and porosity. Furthermore, the scaffold surface should be featured in a way to facilitate cell attachment and proliferation. The hydrophilicity of the scaffold surface is also a facilitator for cell adhesion. Likewise, the porosity of the scaffold allows cell migration and nutrient transference. As mentioned, the physical features of the scaffold are the most important features for the cell growth and proliferation. Besides physical and chemical cues, the mechanical cues are known to be a key player to direct cell fate. In our next research, we are going to focus on the mechanical feature of the cue. In this study, we are planning to use nano-indentation and AFM- based force distribution. According to the results of the planned research, we are going to design a surface patterned scaffold with regulated cross-linking density of the chitin-based scaffold to mimic the natural wings of the butterfly.
- This study ignores using a control scaffold that is commonly used in the medical application of bone fractures that are already available in the market, such as scaffolds based on hydroxyapatite or Bovine Hydroxyapatite. Thus, this data cannot provide a sufficient illustration regarding the advantages of the butterfly wing as a scaffold in bone regenerative.
This study aims to introduce a biomimetic scaffold with appropriate surface topography and chitin-based structure capable of inducing osteogenic differentiation. In our previous reports, we have used herbal derived cellulosic scaffolds to induce osteogenesis and found some clues that increasing the stiffness have improving effect on bone induction. Thus, we tried to find and shortlisted a potential candidate for this biomimicry research considering both chemical structure and surface topological properties. We finally selected butterfly wings and some aquatic insects to evaluate their ability of osteoinduction while the data proved they do not cause any significant local or systemic inflammatory response. In addition to being an innovative and basic research, it is cost-efficient and does not require complex chemical, mechanical or biological processing. We agree with your scientific and applied suggestion that is necessary for its future industrial application. However, comparing the butterfly wing scaffold with the commercially available scaffolds, such as hydroxyapatite or Bovine Hydroxyapatite, requires further in vitro, in vivo and clinical studies that we are planning to assess and address in our future studies. Besides, please be informed that in our project plan, as mentioned earlier, we have already planned and initiated pursuing for a deeper mechanical study to mimic this naturally derived scaffold and evaluate possibility of using it solely or after some modifications for in vivo and clinical applications.
- In the abstract, it was written, "In vivo analysis showed that the scaffold induced no allergic response in Wistar rats." At the same time, the examination carried out was the expression of TNFa and IL6, which are inflammatory markers related to the wound healing process. In contrast, no direct evidence points to "no allergenic" materials such as the determination of IgE expression or levels in experimental animals.
Tumor necrosis factor-alpha (TNFa) is a pleiotropic cytokine that physiologically produces more than one effect in mammalians. TNFa is a major player in inflammatory conditions such as allergenic responses. TNFa effect starts from the beginning of the allergic response (allergen sensitization) and continues to the later stages (activation of inflammatory cascades). Therefore, the expression pattern of TNFa reflects both the inflammatory and allergenic responses. Biologic drugs including anti-TNFa and anti-IL6 are under research to become medically indicated as a treatment for asthma, in which the hypersensitivity and increased response to allergen are the main underlying pathophysiology. However, because factors TNFa and IL6 are indirectly involved in causing allergic responses, we substituted "inflammatory response" for "allergic response" in our manuscript. Also, we will investigate the allergic response by determination of further related cellular and molecular markers in our upcoming papers.
- Can the in vitro study approach conclude that the simply treated P. domeleus butterfly wings can produce biocompatible and biodegradable scaffolds for mesenchymal stem cell culture and proliferation?
Thanks for your accurate review of our manuscript. We agree that further investigation is needed in the case of biocompatibility and biodegradability of the introduced scaffold at the in vivo level. However, the main goal of this study is to examine the wings of P.demoleus butterfly as a potential 3D scaffold for growth and osteogenic differentiation of mesenchymal stem cells. At the same time, the biocompatibility is assessed by MTT assay, DAPI staining and FE-SEM imaging at the cellular level. Mild and then regressive pattern of inflammatory response in rats after transplant of the scaffold also confirms the biocompatibility at the in vivo level. The biodegradability is expected since the main background material of the wings is made up of the chitin, which is a biodegradable natural material. The chitin-based structure of the scaffold has been confirmed by the FTIR spectrum analysis. As earlier mentioned, we are already investigating it (with or without surface modifications) compared to some aquatic insects' wings with different hydrophilicity in vivo with further validation in animal studies and hope that we can publish upcoming data in an independent manuscript.
- Examination of Osteocalcin in mesenchymal stem cell culture was not accompanied by a control group, as shown in Figure 7. Therefore, these data cannot be interpreted ultimately. Likewise, a higher resolution is needed so that visual evaluation can be carried out quickly.
Immunocytochemistry assay has been only done on the scaffold sample, because the RT-qPCR has been recruited formerly to compare the difference between the expression level of Osteocalcin in scaffold and TCPS (control sample). Based on gene expression analysis (RT-qPCR) higher levels of BGLAP (Osteocalcin) gene expression was seen in the scaffold cultures rather control samples on days 14 and 21. Alizarin Red staining has also been used to determine Ca2+ ion deposition, as a fundamental element of mineralized bone ECM. On the other hand, we know that Osteocalcin is a calcium-binding protein. According to the results (figure 5), Ca2+ deposition formation is significantly higher in the scaffold culture rather tissue culture plates.

Reviewer 2 Report
- I would suggest briefly mentioning electrospinning as a mean to prepare ECM-mimicking scaffolds of a variety of materials, e.g. Open Ceramics, 2022, 9, 100237 etc.
- Please increase the resolution of the fonts in all the figures presented.
- Usually it is impossible to measure WCA below 7 Degrees. The authors reported zero. Please provide some details on how it was done.
- Please provide some details on surface energy, if possible which corresponds to specific surface CA.
- 4c, I observe that the vertical axes should be cut to 20% since no useful information above this value is presented.
- The authors discuss the importance of mechanical properties, however, none of them were measured in the study.
Author Response
June 17, 2022
Dear Dr. Prof. Dr. Francesco Puoci
Editor-in-Chief of the Journal of Functional Biomaterials
We would like to express our deep gratitude to you, editors and reviewers for providing us with the insight and direction needed to complete our submitted manuscript under the title of “Crossing phylums: Butterfly wings as a natural perfusable three-dimensional (3D) bio construct for bone tissue engineering”.
We have carefully taken your comments and questions into the consideration for preparing revised manuscript. Please kindly find the revised version of our manuscript in which the revised parts are indicated in red color in revised manuscript. However, to some extent we would like to emphasize the response to your questions and major comments in this reply sheet as follows.
I hope our responses to your comments meet your expectations to initiate its reviewing process. This manuscript has been checked by language editing services prior to submission.
Please kindly let us know if you or reviewer evaluate that this revision needs further revision. We would be very happy if our work could be acceptable for publication in internationally renowned Journal of Functional Biomaterials.
Thanks in advance,
Hosein Shahsavarani, PhD
Reviewer #2
Response to the general comments of reviewer #2: We would like to take this opportunity to express our deep appreciation for your time and comprehensive review of our manuscript. Please find our responses to all your suggestions as follows in advance.
- I would suggest briefly mentioning electrospinning as a mean to prepare ECM-mimicking scaffolds of a variety of materials, e.g. Open Ceramics, 2022, 9, 100237 etc.
Thanks for your suggestion. This electrospinning process has been mentioned in introduction and the paper was cited accordingly.
- Please increase the resolution of the fonts in all the figures presented.
We improved the resolution of all the figures used in the revised version of the manuscript.
- Usually it is impossible to measure WCA below 7 Degrees. The authors reported zero. Please provide some details on how it was done.
We agree that it is usually impossible to report a WCA below 7 degrees. In this study, the butterfly scaffold became hyperhydrophilic after treatments were performed. Figure 2e shows the surface of the scaffold after 5 seconds of the sprinkling water droplet on it, which is completely spread on the scaffold. We explained this part in results to provide better clarity. Also, a video of performing the test was attached to this manuscript.
- Please provide some details on surface energy, if possible, which corresponds to specific surface CA.
In this study, we were only able to examine the water contact angle as a direct analysis of surface chemistry (presence of polar functionalities), which is considered as a required factor for cell adhesion and did not measure surface energy. We were also unable to perform this test during this limited time.
- 4c, I observe that the vertical axes should be cut to 20% since no useful information above this value is presented.
It was corrected in the revised version of manuscript.
- The authors discuss the importance of mechanical properties, however, none of them were measured in the study.
Our main goal in this study is to provide a structurally and topologically suitable scaffold that can provide a suitable surface for cell attachment and differentiation of MSCs to osteoblasts. However, mechanical properties are also essential in tissue engineering and scaffolding. We intend to study the mechanical properties and improve them in future studies. Also, in the abstract and discussion section, where was used the "mechanical" incorrectly, we replaced it with "topographical".
Other changes
Abstract:
Line 25: “topographical” was replaced with “mechanical”.
Line 27,28: “no major inflammatory response” was replaced with “no allergic response”.
Introduction:
Line 44: “MCAs” was deleted.
Line 59: “that” was added.
Line 60: “that” was deleted.
Line 68-70 was added.
Line 73: “eliminating” was replaced with “raising”.
Line 79: “sources” was added.
Line 83-85 was rewritten.
Results:
Section 3.2.4 was rewritten.
Discussion:
Line 63-68 was rewritten.
Line 490: “complete absorption” was replaced with “0â—¦”.
Line 506,507: “no major inflammatory response” was replaced with “no allergic response”.
Line 553,554: “topographical” was replaced with “mechanical”.
Figures:
The resolution of all the figures in the revised version of manuscript was improved.
References: References have been updated.

Round 2
Reviewer 1 Report
The revised version of the manuscript has undergone significant improvements. There are also supported by substituting some more appropriate points to make the sentence clearer. In addition, the author(s) has clarified some issues very well and thoroughly. They have also answered various questions addressed in the previous manuscript.
Furthermore, I recommend accepting this revised version of the manuscript.
Reviewer 2 Report
The manuscript was improved and it can now be accepted.